# In Vitro Properties of Manganese-Substituted Tricalcium Phosphate Coatings for Titanium Biomedical Implants Deposited by Arc Plasma

**DOI:** 10.3390/ma13194411

**Published:** 2020-10-03

**Authors:** Inna V. Fadeeva, Vasilii I. Kalita, Dmitry I. Komlev, Alexei A. Radiuk, Alexander S. Fomin, Galina A. Davidova, Nadezhda K. Fursova, Fadis F. Murzakhanov, Marat R. Gafurov, Marco Fosca, Iulian V. Antoniac, Sergey M. Barinov, Julietta V. Rau

**Affiliations:** 1A.A. Baikov Institute of Metallurgy and Material Science Russian Academy of Sciences, Leninsky, 49, 119334 Moscow, Russian; fadeeva_inna@mail.ru (I.V.F.); vkalita@imet.ac.ru (V.I.K.); imet-lab25@yandex.ru (D.I.K.); imet-lab25@mail.ru (A.A.R.); alex_f81@mail.ru (A.S.F.); barinov_s@mail.ru (S.M.B.); 2Institute of Theoretical and Experimental Biophysics of Russian Academy of Sciences, Institutskaya 3, Puschino, 142290 Moscow, Russian; davidova_g@mail.ru; 3Federal Budget Institution of Science State Scientific Center of Applied Microbiology and Biotechnology, 24 block A, Obolensk, Serpukhov, 142279 Moscow, Russian; n-fursova@yandex.ru; 4Institute of Physics, Kazan Federal University, Kremlevskaya 18, 420008 Kazan, Russian; murzakhanov.fadis@yandex.ru (F.F.M.); mgafurov@gmail.com (M.R.G.); 5Istituto di Struttura della Materia, Consiglio Nazionale delle Ricerche (ISM-CNR), Via del Fosso del Cavaliere 100, 00133 Rome, Italy; marco.fosca@ism.cnr.it; 6Faculty of Materials Science and Engineering, Politehnica University of Bucharest, 313 Splaiul Independentei, District 6, 060042 Bucharest, Romania; antoniac.iulian@gmail.com; 7Department of Analytical, Physical and Colloid Chemistry, Institute of Pharmacy, Sechenov First Moscow State Medical University, Trubetskaya 8, Build. 2, 119991 Moscow, Russian

**Keywords:** manganese, tricalcium phosphate, coatings, titanium implants, arc plasma deposition, electron paramagnetic resonance, mouse fibroblasts, human tooth postnatal dental pulp stem cells

## Abstract

Bioactive manganese (Mn)-doped ceramic coatings for intraosseous titanium (Ti) implants are developed. Arc plasma deposition procedure is used for coatings preparation. X-ray Diffraction, Scanning Electron Microscopy-Energy Dispersive X-ray Spectroscopy, and Electron Paramagnetic Resonance (EPR) methods are applied for coatings characterization. The coatings are homogeneous, composed of the main phase α-tricalcium phosphate (α-TCP) (about 67%) and the minor phase hydroxyapatite (about 33%), and the Mn content is 2.3 wt%. EPR spectroscopy demonstrates that the Mn ions are incorporated in the TCP structure and are present in the coating in Mn^2+^ and Mn^3+^ oxidation states, being aggregated in clusters. The wetting contact angle of the deposited coatings is suitable for cells’ adhesion and proliferation. In vitro soaking in physiological solution for 90 days leads to a drastic change in phase composition; the transformation into calcium carbonate and octacalcium phosphate takes place, and no more Mn is present. The absence of antibacterial activity against *Escherichia coli, Enterococcus faecalis, and Pseudomonas aeruginosa* bacteria strains is observed. A study of the metabolic activity of mouse fibroblasts of the NCTC L929 cell line on the coatings using the MTT (dye compound 3-(4,5-Dimethylthiazol-2-yl)-2,5-diphenyltetrazolium bromide) test demonstrates that there is no toxic effect on the cell culture. Moreover, the coating material supports the adhesion and proliferation of the cells. A good adhesion, spreading, and proliferative activity of the human tooth postnatal dental pulp stem cells (DPSC) is demonstrated. The developed coatings are promising for implant application in orthopedics and dentistry.

## 1. Introduction

The growing number of patients with the pathology of large joints is associated with the aging of the population. Diseases of the musculoskeletal system are one of the reasons for the long-term disability of patients. One of the widely used methods of treating fractures of loaded parts of the skeleton, such as spine, knee, and hip joints, is endoprosthesis. The need for major joint replacement operations is roughly estimated as 200 per 100,000 people. For example, approximately $23 billion was charged for 657,000 primary hip/knee replacement operations in the United States in 2004 [1]. Moreover, prosthetic joint infections occur in about 1.5–2.5% of primary knee and hip arthroplasties [2]. Authors [3] report that dental implant prevalence among US adults significantly increased since 1999 and estimate that by 2026 it could be as high as 23%. This trend provides valuable information for the implant research direction and implant industry development. Emerging new research studies are focused mainly on dental implant surface modification, in order to improve host tissue response and accelerate the healing process. [4] A very important issue for orthopaedic and dental implants is osseointegration. The use of calcium phosphate coatings on metallic implants improves the integration of the implant with the surrounding bone tissue. Plasma deposition has a wide application as a coating technology for intraosseous implants due to the advantage of obtaining coatings with a required porosity and microrelief of the surface [5]. Among the calcium phosphate coatings, the most promising are those based on tricalcium phosphate (Ca_3_(PO_4_)_2_, TCP), characterized by good biocompatibility and resorbability in body fluids [6,7,8]. Indeed, the solubility of TCPs in water is higher than that of hydroxyapatite (HA) [9].

Among the trace elements with therapeutic properties, manganese (Mn) has recently attracted attention for inclusion in various bioceramic materials [10,11,12,13]. Mn is an important micronutrient, being present in various tissues of mammals [14] and fulfilling numerous roles. Mn is involved in bone formation [15], and in antioxidant defence mechanisms preventing the development of malignancies [16]. Mn is a key cofactor of metalloenzymes (oxidases and dehydrogenases), DNA polymerases, and kinases [17]. It was shown to affect integrin activity in human osteoblasts, mediating interaction with the extracellular matrix and being important for cell adhesion, spreading, and proliferation [18,19]. Mn^2+^ ions are also involved in the synthesis of plasma membrane proteins such as collagen, playing a key role in the formation of bones and connective tissues [19]. The effect of Mn on cell functions is dose-dependent, and its release should be carefully regulated. According to authors [19], Mn^2+^ ions exhibit antibacterial activity in concentrations not less than 12.5 µg/mL. Mn ions incorporated in bulk and coating materials are known to inhibit the growth of various bacterial species [11,12,20]. It was found that Mn stimulated the bactericidal activity of polymorphonuclear leukocytes [21].

To our knowledge, authors [13] are the first to report Mn-doped TCP coatings on titanium (Ti) prepared by pulsed laser deposition. The authors also investigated biological properties of their coatings; in particular, the cytotoxicity towards mesenchymal stem cells, reporting an enhanced viability and proliferation of the cells.

In the present work, for the first time, the arc plasma method was used for deposition of Mn-substituted tricalcium phosphate (Mn-TCP) and, for comparison, pure TCP coatings on Ti. The focus of this study was to extend the coatings’ characterization. Their phase composition, morphology, and Mn state and content were reported. Wetting contact angles were measured. In vitro soaking in physiological solution for 90 days was performed. Furthermore, biological properties with two cell lines, mouse fibroblasts NCTC L929 cells, and human tooth postnatal dental pulp stem cells (DPSC) were investigated. The antibacterial activity of the coatings towards the *Escherichia coli* (*E. coli*), *Enterococcus faecalis* (*E. faecalis*), *Pseudomonas aeruginosa* (*P. aeruginosa*) bacteria strains was tested.

## 2. Materials and Methods

Titanium substrates were sprayed with TCP and Mn-TCP using a standard Universal Plasma Unit DC device with a direct current arc plasma torch (Rzhev Elektromekhanika Plant, Rzhev, Russian Federation). The sputtering process was carried out in a chamber with a protective argon atmosphere, the arc current of the plasma torch was 300 A, the voltage was 30 V, and the argon consumption was 40 L/min. Ti wire, containing 0.15% Fe, 0.05% C, 0.08% Si, 0.04% Ni, 0.1% O, and 0.008% H, with a diameter of 2 mm, was used as substrate for the coatings deposition. The Ti substrate rod was rotated at a rate of 160 rpm. Abrasive SiC powder (700 μm particle size) was applied to clean the substrates. 

TCP and Mn-TCP were synthesized by the mechanoactivation method [22]. Calcium oxide (chemical grade, Chimmed, Moscow, Russia), ammonia dihydrogen phosphate (analytical grade, Chimmed, Moscow, Russia), manganese acetate tetrahydrate (analytical grade, Chimmed, Moscow, Russia), and ammonia solution 25% (Chimmed, Moscow, Russia) were used for the syntheses. The reagent quantities were calculated according to the reaction (1):(3−x)CaO + xMn(CH_3_COO)_2_ + 2(NH_4_)_2_HPO_4_ + 2H_2_O → Ca_3−x_Mn_x_(PO_4_)_2_ + 2x CH_3_COONH_4_ + (4−2x)NH_4_OH(1)
where x = 0 or 0.1.

The reagent quantities for x = 0 were as follows: 16.8 g of calcium oxide and 26.4 g of ammonia dihydrogen phosphate. Calcium oxide was heat-treated before synthesis for 1 h at 950 °C. The reagents were placed into a container with 250 g zirconia balls, and then the container was placed in a planetary mill (MP 4/1, “Techno-centr” Ltd., Rybinsk, Russia). The activation was held at 200 rpm for 30 min. For Mn-TCP (x = 0.1), the synthesis was the same as written above with the difference that 2.3 g of manganese acetate tetrahydrate was added into the container. After 30 min, 200 cm^3^ of distilled water was added to both the containers, and grinding continued for another 30 min. The resulting suspension was filtered out, dried in a drying oven (ShS-80, “Smokensk SKTB SPU”, Smolensk, Russia) at 110 °C until completely dry, then subjected to heat treatment at 400 °C to remove the by-products of the synthesis (ammonium acetate and water residues).

The powder obtained after synthesis was pressed in steel molds at a specific pressure of 285 kg/cm^2^ and then sintered at 1200 °C for 4 h. Sintered ceramics was crushed in a planetary mill with zirconia balls for 20 min at 200 rpm, after which a fraction of 30–60 µm was selected using a set of sieves. The particles in the final Mn-TCP ceramic powder had a shape close to spherical, and sizes in the range of 30–60 µm.

The TCP and Mn-TCP ceramics and coatings morphology and elemental composition were investigated by a Scanning Electron Microscope (SEM) (Tescan Vega II, Tescan, Brno, Czech Republic) with Energy Dispersion X-ray analysis-INCA EDX spectrometer.

X-ray Diffraction (XRD) investigations were performed using a Rigaku Ultima IV diffractometer (RIGAKU corporation, Akishima-shi, Tokyo, Japan) with the CuK_α_ radiation in the range of 10–60 (2θ) with a step of 0.02 degree. Phase identification was carried out using the ICDD PDF2 database. To estimate phase ratio, the standard Rietveld refinement method using the Jana 2006 software (Vaclav Petricek, Michal Dusek, JANA98, Institute of Physics, Academy of Sciences of the Czech Republic, Praha, 2000 December) was carried out.

Contact angles of wetting were measured using a sitting drop method. A drop of deionized water with a volume of 2 (±0.1) µL was applied to the surface of the test sample using a syringe dispenser (home-made, Institute of Theoretical and Experimental Biophysics of Russian Academy of Sciences, Puschino, Moscow reg., Russian Federation) and filmed on a stand applying a 5-megapixel video camera PL-A741 (PixeLINK, Rochester, NY, USA), OBJ-11 MMS (Edmund Optics, Barrington, USA) and Fiber-Lite DC-950 (Dolan-Jenner Industries, Boxborough, USA), which provides uniform illumination. The resulting drop image was processed using the Low Bond Axisymmetric Drop Shape Analysis method by the DS LB-ADSA module to the ImageJ application, developed by Biomedical Imaging Group EPFL (Lausanne, Switzerland), which allows one to get the desired value of the wetting edge angle. As control, standard glass slides were used. Five measurements were made for each sample and averaged to compare the treated and untreated surfaces. The Mann–Whitney U-test was applied to calculate the statistical significance of differences between the groups.

Conventional Electron Paramagnetic Resonance (EPR) measurements were carried out on ESP-300 spectrometer (Bruker, Germany) operating at X-band (9–10 GHz) microwave range at room temperature using various cavities. To check for potential anion impurities, the samples were irradiated by X-ray at room temperature during 1 h with the estimated dose of 20 kGy.

Soaking procedure in physiological solution was carried out as follows: Ti substrates coated with Mn-TCP were placed into a container with physiological solution containing TRIS-buffer (Chimmed, Russia) at pH 7.4. The container was stored in a thermostat (TSO 1/80 SPU, “Smokensk SKTB SPU”, Smolensk, Russia) at 37 °C for 90 days. After soaking, the samples were removed from the solution, washed with distilled water, and air-dried.

The cytotoxicity of the samples was evaluated using extracts from materials in accordance with the requirements of ISO 10993.5-2011 in Dulbecco’s Modified Eagle’s culture medium (DMEM) by MTT test. To obtain extracts, coated Ti samples were incubated in DMEM for 3 days at 37 °C. All the materials were placed in Petri dishes and sterilized for 1 h at 180 °С. The DMEM supplemented with 100 unit/mL of penicillin/streptomycin was used as a model medium for the preparation of extracts. The preparation of extracts was carried out for 3 days at 37 °C, following the aseptic conditions. Three samples were made for each material. The ratio of the surface area of the material to the volume of the DMEM medium was 3 cm^2^/mL. In the experiments, mouse fibroblasts of the NCTC L929 line were used, the seeding cell concentration was 30,000/cm^2^. A total of 24 h after plating the cells into the wells of a 96-well plate, the culture medium was replaced with extracts and cultured at 37 °C for 24 h. After the end of cultivation, the cell viability was assessed by the MTT test.

Human tooth postnatal DPSCs were isolated from the rudiment of the third molar extracted for orthodontic indications, as described earlier in [23]. The cells were grown in the DMEM medium with the addition of 10% Fetal Bovine Serum (FBS, HyClone) in a humidified incubator (Binder, Tuttlingen, Germany) at 37 °C and 5% CO_2_. In the primary cell culture, the medium was changed every 24 h. The cells were maintained until dense islets or monolayer cells were formed, and then cultivated for growth. For this study, cells from the third and fourth passages were used.

Cell viability on the surface of the studied samples was assessed using the AlamarBlue assay [24] by means of resazurin (7-hydroxy-3H-phenoxazine-3-on-10-sodium salt oxide) (Sigma-Aldrich), which is reduced by enzymes of living cells to the fluorescent product resarufin. DPSCs were seeded on the surface of the Mn-TCP coated Ti samples at a concentration of 30,000 cells/cm^2^ in the DMEM medium in 24-well plate, and glass slides of the same size were used as controls. After 24, 48, and 168 h, the samples were transferred to the wells of a 24-well plate and 500 µL of resazurin was added to each well with a final concentration of 50 mmol/L. The plates were incubated for 3 h. The fluorescence of the reduced dye was determined using a spectrofluorometer Tecan Infinity camera F200 (Tecan, Männedorf, Switzerland) (λ_excitation_ = 530 nm, λ_emission_ = 590 nm).

A transgenic culture of DPSCs carrying the green fluorescent protein gene (GFP-DPSC) was prepared using the LVT-TagGFP2 lentivector (Eurogene, Moscow, Russia). The cells were transduced according to [25]. DPSCs of the second passage were sown in a 24-well plate in a concentration of 10^4^ cells/well. A day after seeding, lentiviral vectors were added to the culture medium; a day later, the culture medium was changed. On the third day after adding the lentiviral vectors, the development of the GFP expression by the level of fluorescence was observed in cells using a Axiovert 200 fluorescent microscope (Carl Zeiss, Oberkochen, Germany) [26]. The resulting cell culture (GFP-DPSC) was used to determine the adhesion characteristics of the surfaces of TCP and Mn-TCP coatings and their ability to support cell proliferation by fluorescence microscopy. GFP-DPSCs were seeded on the surface of the coatings (n = 3 per study) at a concentration of 30,000 cells/cm^2^ (DMEM + 10% FBS) in 24-well plates, and glass slides of the same size were used as controls. At day 1(24 h), day 2(48 h), and day 7(168 h) after seeding, the cells were visualized on the surface of the materials using an Axiovert 200 fluorescence microscope with λ_excitation_ = 450–490 nm and λ_emission_ = 515–565 nm, having previously stained the dead cells with propidium iodide λ_excitation_ = 450–490 and λ_emission_ = 515–565 nm.

Antibacterial activity was tested on the *E. coli*, *E. faecalis*, and *P. aeruginosa* bacteria strains. Bacteria were grown in a dense nutrient medium Mueller Hinton Agar (HiMedia, India). The antibacterial activity was determined by applying the active material to a freshly seeded bacterial lawn. The culture of the strains was cultivated from a concentration of 10^6^ CFU (colony forming units)/mL (100 µL per dish). Rubbed with a sterile cotton swab, sample plates were placed on the lawn for 10 min. Petri dishes with lawn and test samples were placed in a thermostat and cultured for 18–24 h. The antibacterial activity was evaluated based on the presence and width of bacterial culture growth suppression zone.

Ethics statement. Normal fibroblast cell line NCTC L929 of murine subcutaneous connective tissue was obtained from the Russian collection of cell cultures (Institute of Cytology of the Russian Academy of Sciences (Moscow)). The DPSC culture was obtained from freshly extracted third molar teeth (donor age 16 years) with a formed root of at least two-thirds, which were extracted for orthodontic reasons at the Central Research Institute of Dentistry and Maxillofacial Surgery of the Ministry of Health of Russia (Moscow) by agreement with the ethics committee after the consent signed by the patient’s parents. All experiments were performed in accordance with the principle of good clinical practice and ethical principles contained in the current edition of the Declaration of Helsinki.

## 3. Results and Discussion

According to the Rietveld refinement data, the Mn-TCP powder consisted of two phases β-Ca_3_(PO_4_)_2_ (about 70%) and HA (about 30%), while the composition of the deposited coatings slightly changed: α-TCP (about 67%) and HA (about 33%). The XRD spectra of the initial TCP and Mn-TCP powders and deposited TCP and Mn-TCP coatings are presented in Figure 1. The reason of α-TCP phase presence in the coatings can be likely justified by the deposition conditions, i.e., high temperature of plasma followed by a fast cooling down of the coating on the Ti substrate. As to the powders, it is probable that Mn incorporation into the TCP structure stabilized the β-TCP phase. This hypothesis can be supported by the ref. [27]. Authors [27] found that the increase of Mn concentration leads to the β-TCP phase formation, while α-TCP is formed at low Mn content. 

Mn content in the powder and in the coating was determined by the SEM-EDX analysis as 2.5 and 2.3 wt%, respectively. The elemental composition of materials is presented in Table 1.

No resonance signals were observed in the undoped TCP powder sample by EPR. This means that the concentration of impurity paramagnetic centers (even if they exist) is less than 10^14^ spins per gram as measured at room temperature [28]. Room temperature EPR spectra for the Mn-TCP powder characterized by a typical intense signal in the magnetic field of 300–400 mT are presented in Figure 2. The obtained data are in accordance with the ref. [29].

Generally, the EPR spectra of Mn^2+^-containing compounds are sufficiently complex, especially in low-symmetry polycrystalline systems, as the averaging across all possible single crystal orientations, the overlap of signals from multiple possible sites, spatial inhomogeneity of the crystal structure, and interaction between the electronic spins at their high concentrations lead to the broadening of the spectra [30]. The spectra at low manganese amount (x < 0.01) are mainly defined by typical Mn^2+^ EPR spectral features—zero field splitting (ZFS) of the ^6^S_5/2_ ground state electron spin *S* = 5/2, and hyperfine coupling to the 100% abundant ^55^Mn nuclear spin with *I* = 5/2. 

Conventionally, the Mn^2+^-TCP systems are described by the spin-Hamiltonian (*Ĥ*) [30]:*Ĥ* = *gβBS* + *D*[*S*^2^_*z*_ − *S*(*S* + 1)/3] + *E*(*S*^2^_*x*_ − *S*^2^_*y*_) + *ASI*(2)
where *S*_*x*,*y*,*z*_ are the projections of the electronic spin *S* = 5/2 on the principal direction of *D*; *D* and *E* are the commonly used (axial and orthorhombic, correspondingly) parameters for ZFS; *g* and *A* are the isotropic *g*-factor and the hyperfine constant, respectively. In previous studies [27,31], it was shown for Mn-doped HA and TCP that *g* ≈ 2.001, *D* = 53(4) mT (1.5 GHz), *Е* = 14(3) mT (0.4 GHz), and *A* ≈ 9.4(4) mT. Our results (Figure 3) can be satisfactorily described by *g* ≈ 2.003, *D* = 42(5) mT, *Е* = 9(2) mT, and *A* ≈ 9.5(5) mT. From the analysis of the high-frequency (W-band, 94 GHz) EPR results, it was concluded that even at low manganese concentrations, Mn^2+^ ions substitute not only Ca(5) site, as it was predicted by calculations [32], but at least three of the five possible calcium sites, presumably Ca(5), Ca(4), and Ca(2), in TCP structure, and both possible sites in the HA structure [12].

With increasing the Mn^2+^ concentration, the average distance between paramagnetic ions decreases, and the rising contribution of electronic spin-spin interaction leads to the broadening of the EPR lines and disappearance of the resolved hyperfine structure. Further increase of manganese amount is followed by the EPR spectrum narrowing, due to the exchange interaction. From the comparison of the obtained EPR pattern (Figure 3) with the results presented in [12], one can give the estimation of the Mn^2+^ amount in the investigated powder as higher than 2 wt%, which is in good agreement with the results obtained by the EDX analysis.

One of the approaches for investigation of the EPR-silent materials is the application of the ionizing radiation, like X-ray, to create stable radiation-induced paramagnetic centers [30]. Various radiation-induced paramagnetic species (mainly anion radicals) have been identified in HA and TCP materials. Among them, oxygen radicals $O^{-}$, trapped atomic hydrogen centers and holes, carbonate radicals ${\mathrm{CO}}_{2}^{-}$, ${\mathrm{CO}}_{3}^{-}$, ${\mathrm{CO}}_{3}^{3-}$, and color centers were observed. In a number of publications [28,30], we demonstrated a high sensitivity of EPR methods to detect the radiation-induced nitrate radicals and suitability of ${\mathrm{NO}}_{3}^{2-}$ to serve as an effective spin probe for studying the structure and chemical composition of HAs. Comparing to the Mn^2+^ signal (see Figure 3), no other impurity radiation-induced signals were detected within the accuracy limit of our measurements (neither in powder nor on the deposited samples) that additionally confirms high purity of the matrices studied. 

Figure 4 shows the EPR spectra from the deposited samples. A broad, intensive EPR line with the linewidth of 200–300 mT and effective g-factor of 2.4–3.6 (depending on the orientation of the external magnetic field to the substrate plane) was detected for the Mn-containing sample. The position of the EPR line is unusual for Mn^2+^ in (nano)crystals and powders. Assuming that the obtained EPR spectrum is still due to the manganese complexes, one may suppose the existence of manganese clusters (for example, in the form of Mn_2_O_3_) and/or high-spin manganese systems (like Mn^3+^) on the substrate surface.

Integer high-spin state (S = 2), as for example for the most studied case of Mn^3+^ ion in superoxide dismutase (SOD), is known to be centered at an effective g-value of 8.17 [34]. i.e., obtained in the low magnetic fields of 40–120 mT. It should be noted that 8.17 is an effective g-factor. From Equation (1) in [34], the following spin-Hamiltonian parameters for Mn^3+^ in Mn-containing superoxide dismutase were extracted: D = 63 GHz, E= 7.2 GHz, g = 2.00. Due to the large D (ZFS), the conventional perpendicular polarization of the allowed EPR transitions with Ms = ±1 are not possible at X-band microwave frequencies. However, EPR signals from integer spin systems can be detected when the oscillating magnetic field, applied to induce a spin-state transition, is oriented parallel to the B. The parallel magnetic field orientation allows transitions between the Ms = ±2 energy levels [27]. Indeed, in parallel mode, in the magnetic fields of less than 200 mT, a broad signal was detected in the Mn-TCP deposited coating (Figure 5).

The morphology of the coatings was investigated by SEM. The prepared TCP coatings (Figure 6A,B) are homogeneous and non-compact due to the presence of small pores. The Mn-TCP coatings’ surface (Figure 6C,D) is also rough with some crystalline inclusions.

Soaking in physiological solution at 37 °C for 90 days induced drastic changes in the TCP and Mn-TCP coatings (see Figure 7). 

The morphological changes in the coating can be well appreciated from the comparison between Figure 6 and Figure 7. The SEM-EDX elemental analysis of the coatings composition was performed, and the following relation of Ca:P:C:O was detected: 1:0:1.1:3.1, corresponding to calcium carbonate (CaCO_3_). The transformation of calcium phosphate into calcium carbonate may take place according to the following equation of TCP dissolution (3):Ca_3_(PO_4_)_2_ + 3CO_2_ + 3H_2_O → 3CaCO_3_ + 2HPO_4_^2−^ + 4H^+^(3)

Furthermore, the surface morphology was characterized by the presence on the CaCO_3_ surface of the groups of lamellar crystals assembled in “rosettes”. Such morphology resembles the octacalcium phosphate (OCP) crystals described in [35,36,37]. The assumption about the OCP formation was confirmed by the EDX analysis of the crystals. The Ca/P ratio was determined to be 1.3, corresponding to OCP. OCP crystals are formed in the coating after soaking in physiological solution in accordance with the following transformation of the α-TCP (4):3Ca_3_(PO_4_)_2_ + 7H_2_O → Ca_8_H_2_(PO_4_)_6_·5H_2_O + Ca(OH)_2_(4)

Since α-TCP phase was detected in the coating, it can be assumed that at 37 °C and at pH close to 7, OCP is formed in the coating in aqueous medium. Previously, the conversion of TCP to OCP was studied in [38], where it was shown that under such conditions (hydrolysis at pH 7.4 and 37 °C), OCP crystals are formed. 

According to the EDX analysis, no Mn is detected in the coating after 90 days of soaking in the physiological solution, likely because all Mn ions have been transferred in the solution. This fact, and the morphological changes, confirm that the deposited coatings are bioresorbable. The coating’s resorption can be represented by the Equation (5):Ca_2.9_Mn_0.1_(PO_4_)_2_ + 3CO_2_ + 3H_2_O → 2.9CaCO_3_ + 2HPO_4_^2−^ + 0.1Mn^2+^ + 4H^+^ + 0.1CO_3_^2−^(5)

Good wetting of the implant surface with physiological fluids and blood is a necessary condition for their good biointegration. The wettability of the surface, which is a physical consequence of the specific chemical structure and physical nature of the surface, affects the adhesion and spreading of cells on the surface of materials through the adsorption of adhesive proteins of blood serum [39]. The measurement of edge contact angles is an important assessment of the properties of a biomaterial [40]. Cell adhesion and spreading are most pronounced on moderately hydrophobic surfaces (40–80°), while hydrophobic or non-ionic hydrophilic surfaces are less suitable for cells attachment [40]. The measurement of the wetting contact angles, presented in Figure 8, showed a significant change in the wettability of the surface of Mn-TCP coated Ti samples compared to the TCP coated Ti. The average value (n = 10) of the edge wetting angle of the surface decreased from 90 ± 2° for the TCP surface to 70 ± 5° for the Mn-TCP surface. Therefore, it can be concluded that the introduction of Mn leads to hydrophilization of the TCP surface.

The MTT test of metabolic activity of the NCTC L929 mouse fibroblasts during incubation for 24 h with extracts from TCP and Mn-TCP materials was carried out. The results are shown in Figure 9. As can be seen from the figure, the metabolic activity of NCTC L929 fibroblasts both for TCP and for Mn-TCP is very close to the control sample and, therefore, no cytotoxicity is registered.

A transgenic DPSC culture carrying the fluorescent green protein gene was used to study the adhesion, spreading, and proliferative activity of cells on the surface of the coating materials. This allowed us to visualize the cells in the process of cultivation on the material without the use of fluorescent dyes and to study their dynamics on the material’s surface. The results of the study of adhesion and growth of cells labeled with GFP are shown in micrographs in Figure 10.

The proliferation activity was evaluated using the growth curve method (Figure 11).

The normal morphology of GFP-DPSCs is observed on all the studied samples, but the density of the cell layer on the studied samples is slightly lower than in the control (glass cover), which is probably due to the fact that the glass is a more hydrophilic material compared to Mn-TCP, and the cell adhesion to its surface is better (53° compared to 70°). Despite this, on the surface of samples a large number of cells were present, the cells were viable and homogeneously spread, indicating no cytotoxic effects of the coating materials. At the same time, the number of cells on Ti coated with Mn-TCP is significantly higher than on Ti coated with TCP. An increase in the number of cells on the surface of all samples on the second and seventh days indicates the proliferative activity of cells cultured on these materials.

One of the most promising methods for determining the viability and proliferative activity of cells seeded on the surface of materials is the resazurin test [41], because the fluorescent signal in the resazurin test is directly related to the number of cells. From the analysis of changes in the fluorescent signal at different stages of DPSC cultured on the surface of materials, we can say that all samples have the greatest increase in cell number between 48 and 168 h of cultivation (see Figure 12). At the same time, after 48 and 168 h, the number of cells on the surface of the Mn-TCP sample is higher (*p* < 0.01) than on the sample that does not contain Mn, i.e., TCP. These results coincide with the data of the evaluation of proliferative activity by direct cell counting (see Figure 11). It is known that Mn^2+^ ions strongly affect the affinity of integrin to ligands and, consequently, the adhesion of cells to extracellular matrix proteins [19]. This explains our results of improved adhesion and proliferation of DPSCs on coatings containing manganese.

The adhesion characteristics of the TCP and Mn-TCP surfaces, and their ability to support the DPSCs proliferation, were visualized by SEM. For TCP, the results are presented in Figure 13A,B and for Mn-TCP - in Figure 13C,D. It can be observed that the cells are well spread on the surface, which indicates its good matrix properties. The growth of cell mass on the surface of the Mn-TCP coating occurs faster compared to the TCP coating’s surface, which is likely due to the influence of the manganese. This observation is confirmed by our previous studies regarding the Mn-TCP powders with different concentrations of Mn [12]. It was found that Mn ions have a positive effect on adipose-derived mesenchymal stem cells’ viability and proliferation, demonstrating no inhibitory or adverse effects, and on cells differentiation into osteogenic, adipogenic, and chondrogenic lineages.

The antibacterial activity of Mn-TCP powders, containing various amount of Mn, was studied in our previous work [12]. According to the results obtained in [12], TCP doped with the highest concentration of Mn, corresponding to the one used in this work, inhibited the growth of the tested bacterial species: *Staphylococcus aureus*, *Salmonella typhi*, *E. coli*, *E. faecalis*, and *P. aeruginosa*. In this work, we tested the antibacterial activity of the prepared TCP and Mn-TCP coatings on Ti against *E. coli*, *E.*
*faecalis*, and *P. aeruginosa* strains, and the results are presented in Figure 14. As can be observed from Figure 14, no bacteria suppression zone can be distinguished.

In detail, the results obtained in [12] evidence that the Mn-TCP powder inhibition of *E. coli* was about 7%, of *E. faecalis* about 20%, and of *P. aeruginosa* about 15%, whereas the TCP powder inhibition of the mentioned bacteria was only a few%. It should be pointed out that in that work, the growth of microorganisms was assessed by reading the optical density of bacteria strains on a biophotometer (Eppendorf, Germany), which is a more precise method. Concerning the coatings deposited in this work, the lack of antibacterial properties of Mn-TCP against all the three investigated strains can likely be explained by the Mn state in the coating, since according to the EPR analysis, Mn is present in the form of clusters, which are likely also less soluble.

## 4. Conclusions

The Mn-TCP coatings on Ti were prepared by arc plasma spraying. The coatings are composed of α-Ca_3_(PO_4_)_2_ (about 67%) and HA (33%). The microstructure of Mn-TCP coating is homogeneous and non-compact due to the presence of some small pores. The Mn content in the coatings is 2.3 wt%. Manganese is present in the coating as Mn^2+^ and Mn^3+^ clusters, non-homogeneously distributed within the coating. The wettability of the coatings is suitable for cells attachment and proliferation. In vitro soaking of Mn-TCP coating in physiological solution for 90 days leads to the change in its phase composition. It is composed of calcium carbonate and octacalcium phosphate crystallites, and Mn is no more present in the coating.

The lack of antibacterial properties of Mn-TCP coated Ti against *E.*
*coli*, *E.*
*faecium*, and *P.*
*aeruginosa* was observed. Both the investigated cell lines, the NCTC L929 mouse fibroblasts, and the human tooth postnatal dental pulp stem cells seeded on the coating’s surface exhibited good adhesion, viability, spreading, and proliferation characteristics.

The developed coatings are recommended for biomedical Ti implant applications in orthopaedics and dentistry.

## Figures and Tables

**Figure 1 materials-13-04411-f001:**
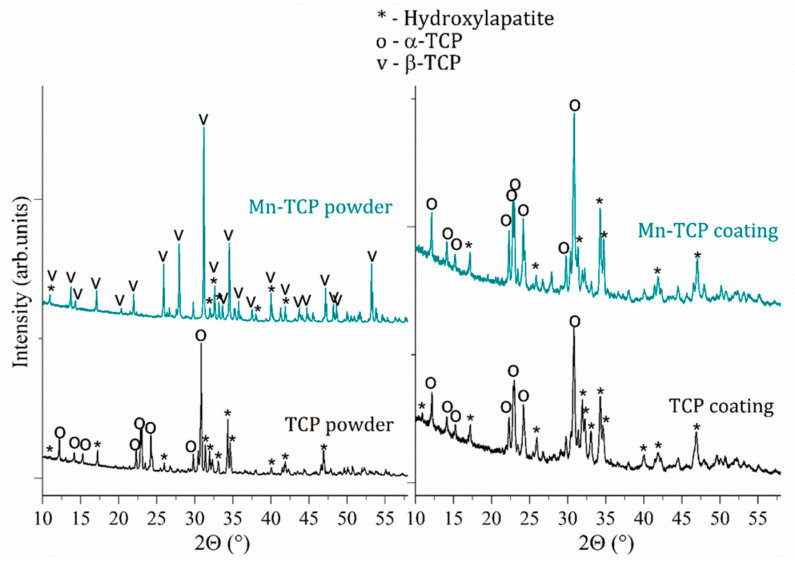
XRD spectra of TCP and Mn-TCP powders and of TCP and Mn-TCP coatings.

**Figure 2 materials-13-04411-f002:**
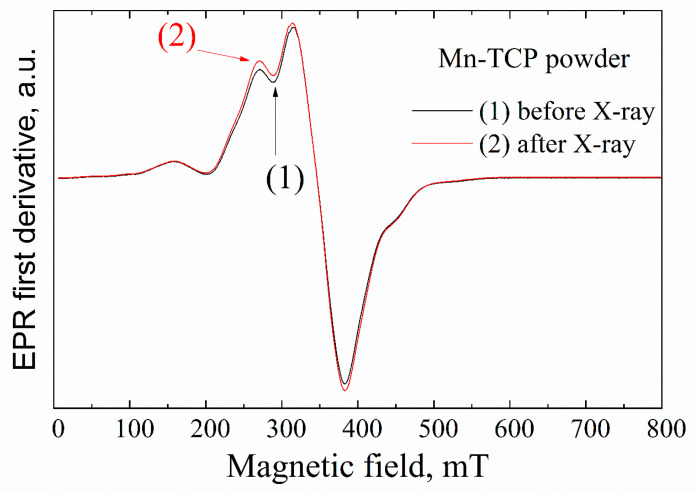
EPR spectra of Mn-TCP powder and X-irradiated Mn-TCP powder.

**Figure 3 materials-13-04411-f003:**
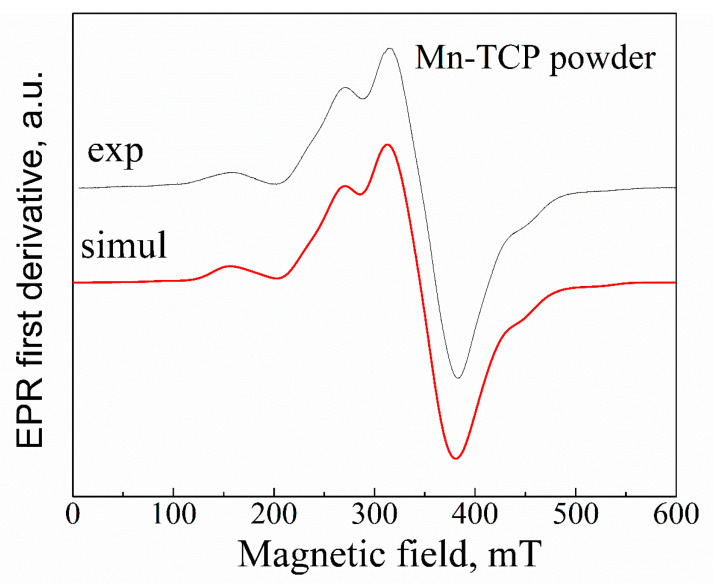
Experimental (exp) and simulated (simul) EPR spectra. Parameters for simulation are given in the text. Simulation is done by the EasySpin utility [33].

**Figure 4 materials-13-04411-f004:**
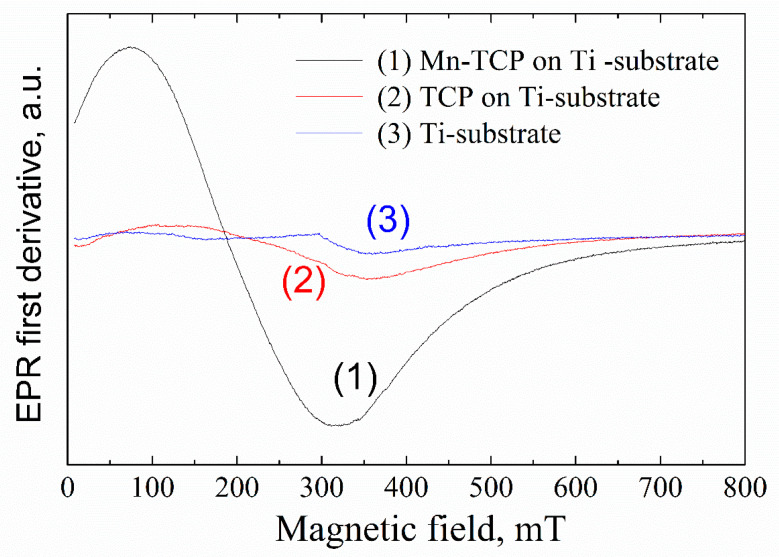
EPR of the deposited Mn-TCP sample (black curve) compared to the TCP-deposited (red) and pure Ti substrate.

**Figure 5 materials-13-04411-f005:**
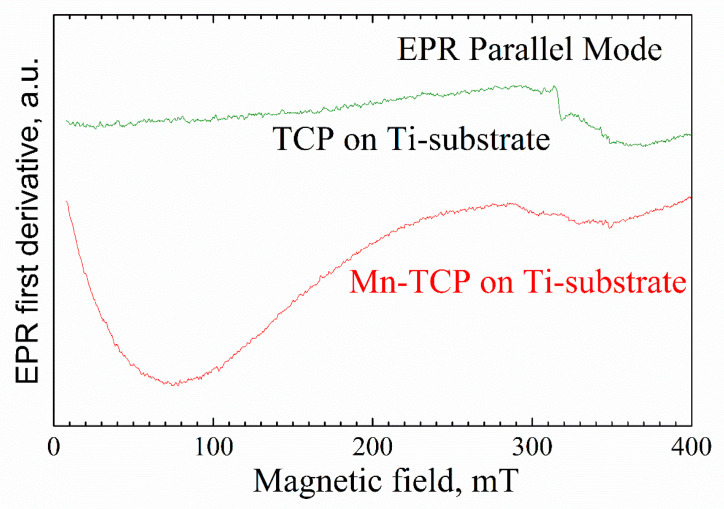
EPR of the deposited species in the parallel mode of the EPR dual mode cavity ER 4116DM.

**Figure 6 materials-13-04411-f006:**
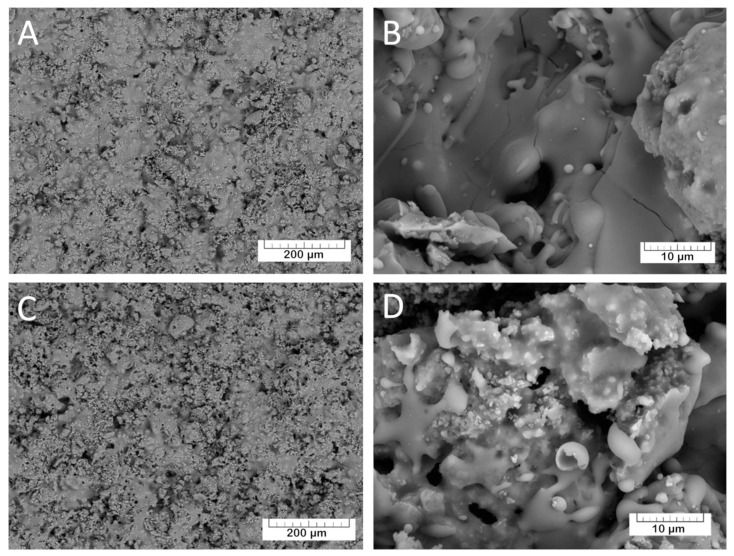
Morphology of the TCP (**A**,**B**) and Mn-TCP coatings (**C**,**D**) at different magnification.

**Figure 7 materials-13-04411-f007:**
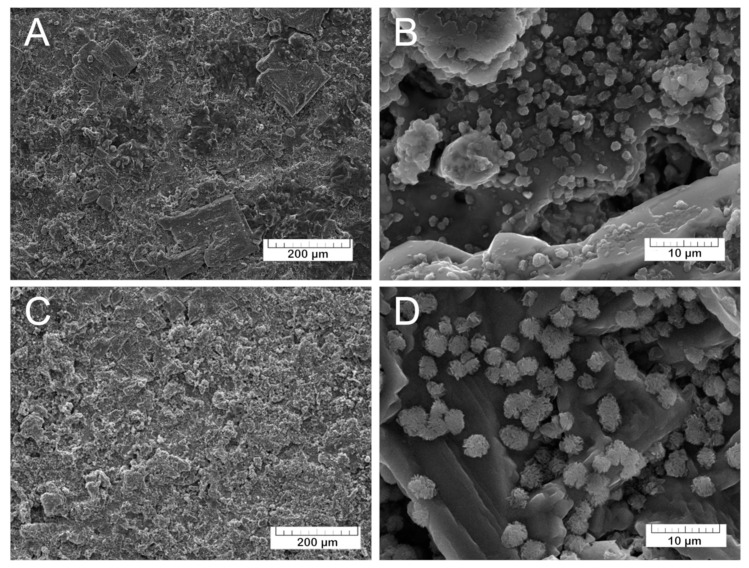
The morphology of the TCP coating (**A**,**B**) and Mn-TCP coating (**C**,**D**) after soaking in physiological solution for 90 days at various magnifications.

**Figure 8 materials-13-04411-f008:**
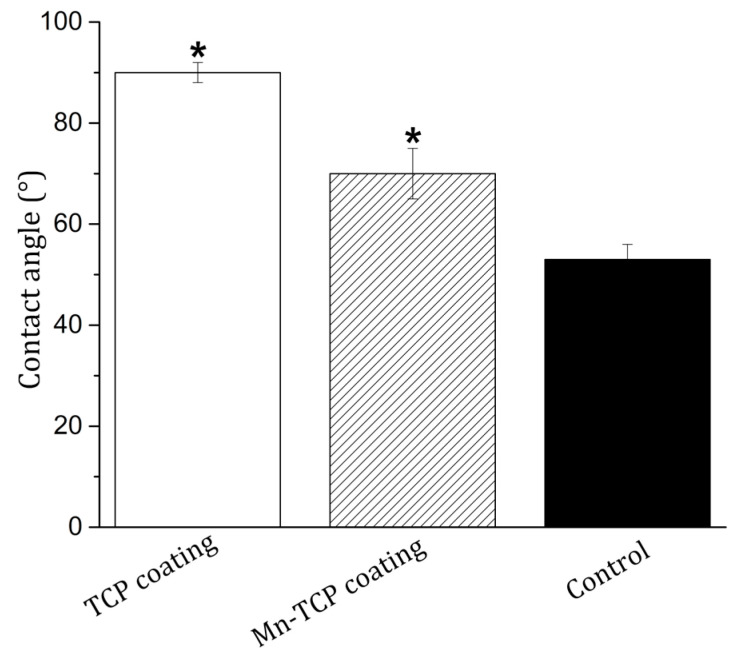
Contact angles of wetting of the surface of TCP and Mn-TCP coated Ti samples with distilled water (*, *p* < 0.01). Control sample–glass slide. The error bar is Mean ± SD.

**Figure 9 materials-13-04411-f009:**
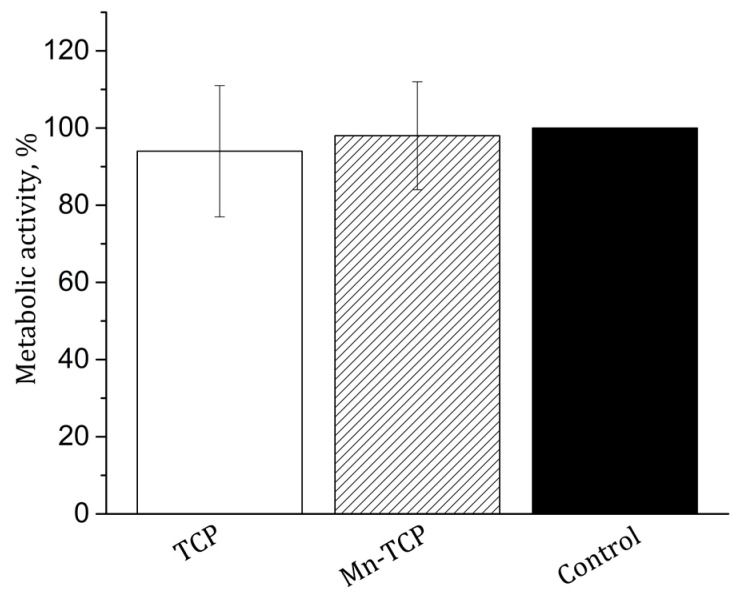
MTT test. Metabolic activity of the NCTC L929 cells during incubation for 24 h with extracts from TCP and Mn-TCP materials, normalized to the control cells cultured in the absence of the mentioned materials. The error bar is Mean ± SD.

**Figure 10 materials-13-04411-f010:**
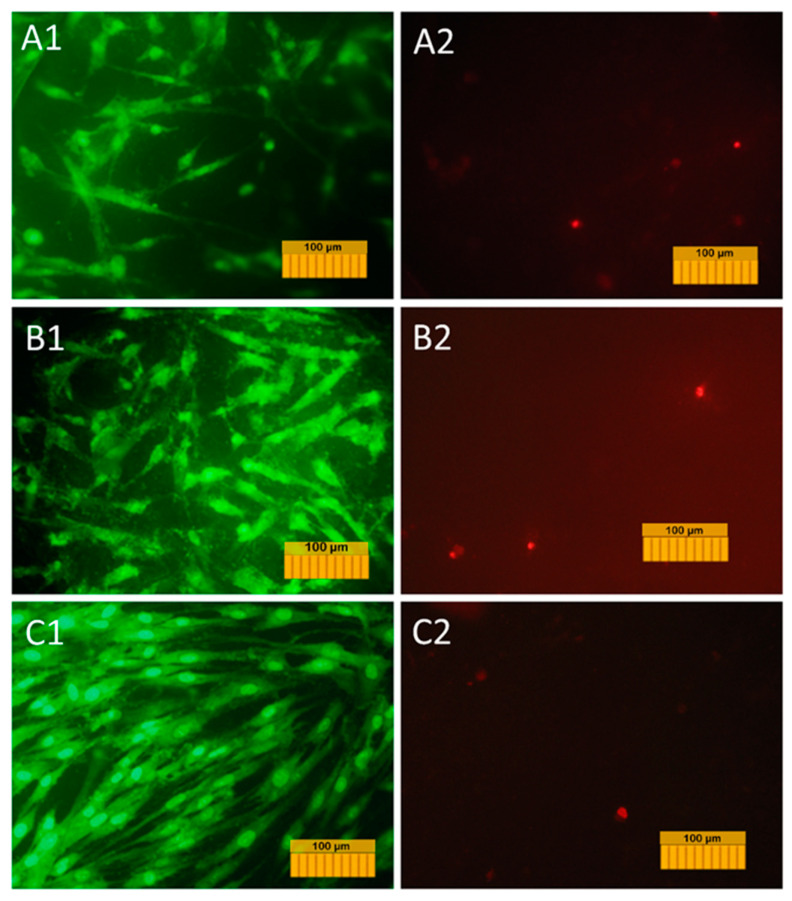
Micrographs of transformed GFP-DPSCs cultured on the surface of samples 48 h after seeding: **A**—TCP coating on Ti; **B**—Mn-TCP coating on Ti, **C**—control sample (1—alive cells with intrinsic green fluorescence; 2—dead cells marked with propidium iodide).

**Figure 11 materials-13-04411-f011:**
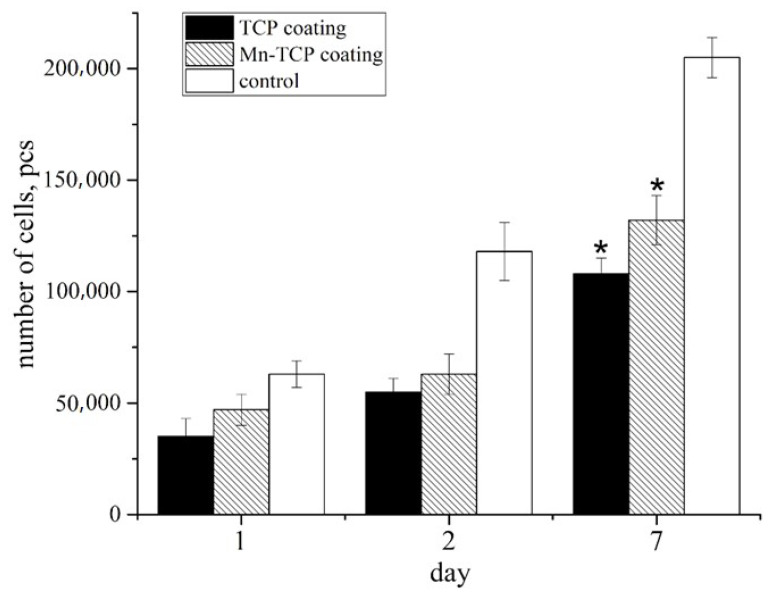
The number of DPSCs seeded on the surface of TCP and Mn-TCP coatings and on the control sample (*, *p* < 0.01). The error bar is Mean ± SD.

**Figure 12 materials-13-04411-f012:**
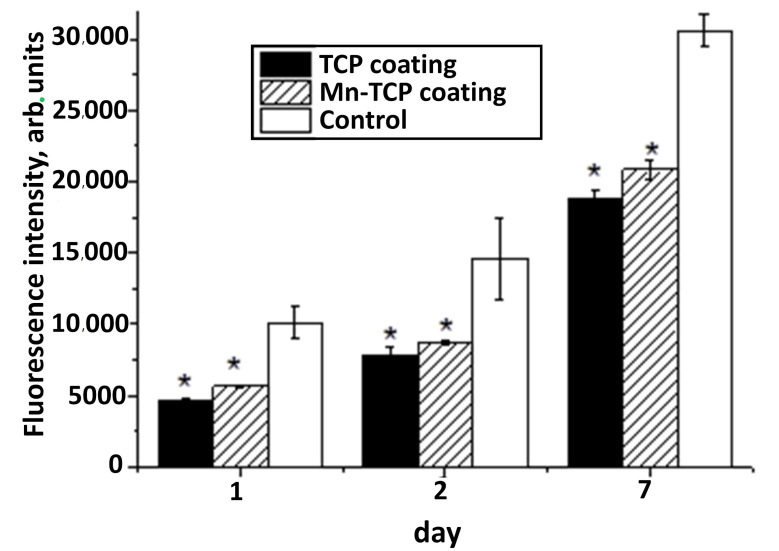
The fluorescence intensity of DPSCs seeded on the surface of TCP and Mn-TCP coatings and on the control sample (*, *p* < 0.01). The error bar is Mean ± SD.

**Figure 13 materials-13-04411-f013:**
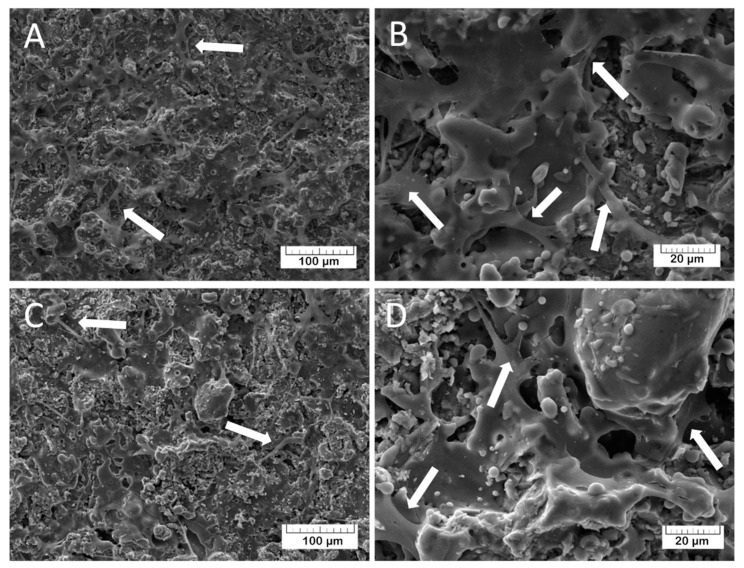
SEM images of DPSCs on the coatings after 2 days of seeding at different magnification: (**A**,**B**) TCP; (**C**,**D**) Mn-TCP. DPSC pseudopods are highlighted by arrows.

**Figure 14 materials-13-04411-f014:**
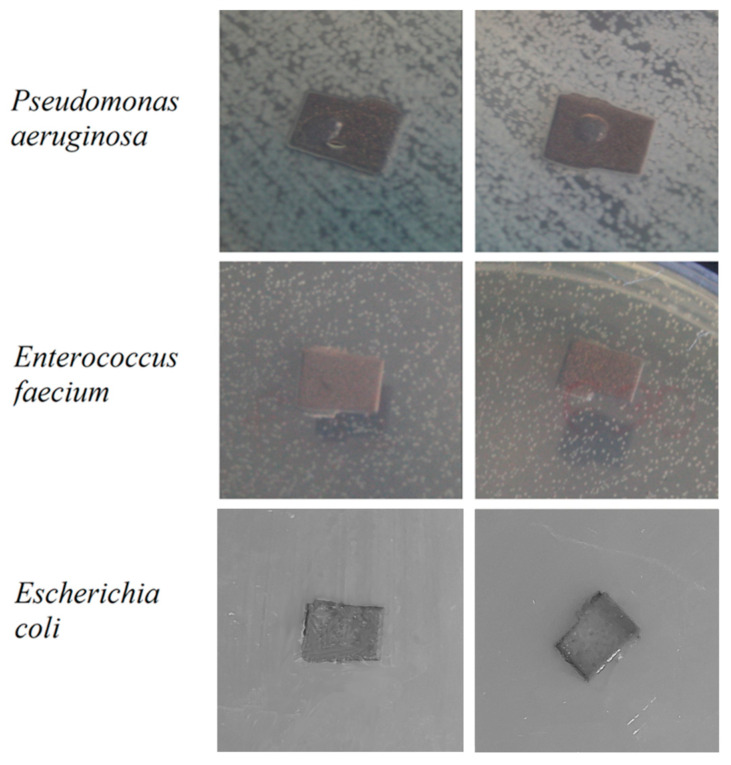
Ti substrates coated with TCP (samples on the left) and Mn-TCP (samples on the right) with *E.*
*coli*, *E.*
*faecium*, *P.*
*aeruginosa*.

**Table 1 materials-13-04411-t001:** SEM-EDX analysis of the Mn-TCP powder and coating.

Powder/Element	O	P	Ca	Mn	Total
Weight %	40.44	19.97	37.07	2.52	100.00
Atomic %	61.34	15.33	22.22	1.11	100.00
Coating/Element	O	P	Ca	Mn	Total
Weight %	40.99	17.25	39.47	2.29	100.00
Atomic %	61.81	13.43	23.75	1.01	100.00

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
