# Peer review of "In Vitro Properties of Manganese-Substituted Tricalcium Phosphate Coatings for Titanium Biomedical Implants Deposited by Arc Plasma"

_materials, 2020, doi:10.3390/ma13194411_

Round 1

Reviewer 1 Report

In this manuscript, the authors deposited biomedical implants with Mn-Substituted TCP coating by arc plasma. Detailed characterisation of the coating were shown and the cell culture behaviour and bacteria activity are presented. 

It is recommended to publish this manuscript after the following concerns are addressed:

(1) In abstract, please show full name of abbrivatio when they are first appeared. such as "TCP". Present tense should be used in the abstract?

(2) "To our knowledge, the only literature reference reporting Mn doped TCP coatings on titanium(Ti) prepared by pulsed laser deposition is [9]." should be improved for the writing. should be "...laser deposition was carried by authors [Ref]". and I think authors in [9] are pioneers in doing this kind of research but not the only one. Please see the cited papers of [9].

(3) in Figure 8 and some other figures ."the error bar is +_SD" should be "the error bar is Mean +_SD"

(4) please explain why antibacterial activity disappear and how to recover the antibacterial activity, since  it is important for implants

(5) where is the DPCs in figure 12? I do not see cells on SEM

(6) Resolution of bacteria suppression pictures shovel be improved

Reviewer 2 Report

In this study, the authors characterize TCP and Mn-TCP coatings on Ti substrates as a potential material used in implant applications. A suggestion below should be considered for improving the quality of the article.

  1. After the sentence, there is a need od citation. “Indeed, the solubility of TCPs in water is higher than that of hydroxyapatite (HA).”
  2. Lines 175-176, there is a mistake. Does Mn-TCP powder consist phase β-Ca3(PO4)2 instead of α-Ca3(PO4)2?
  3. It is not really clear for the reviewer why the authors measure EPR spectra for samples before and after X-ray?
  4. Figures 2 and 3 need the full name of x and y-axes.
  5. Figures 6A and 6C should have the same magnification.
  6. In the materials and methods section, there is a lack of description of the sample soaking in the physiological solution procedure.
  7. Figure 8 presents contact angles of wetting of the surface. Y-ax is described as ‘Contact angle’ expressed as a percentage, is it a mistake or is it relative value? If it is a relative value, it should be clarified in the text. Could the authors show also the significance for p<0.05?
  8. Figure 8 and Figure 9 could have consistently the control sample at the first place or last place.
  9. Could the authors specify how they obtain extracts from the materials? Are they obtain from powders or coatings, and how long the materials were incubated in DMEM?
  10. Could the authors show also the significance for p<0.05? In figures 8, 9 and 11.

Reviewer 3 Report

The manuscript of Fadeeva et al. reports on the development of manganese-doped coatings, prepared by arc plasma deposition on the titanium surface. The authors describe the preparation of coating and investigate its physical, chemical and biological properties in vitro. The authors suggest that this coating can be used for the medical titanium implants to improve their biocompatibility.
The article employs a wide range of methods for the coating characterization, starting from the basic physical and chemical properties and moving all the way to biocompatibility assays. It is an interesting and comprehensive study, that certainly deserves to be published. However, I would like to make a few suggestions that, in my opinion, could improve the manuscript.

In the introduction, the authors describe the growing need for endoprostheses, particularly those placed at the large joints. At the same time, dental surgery is among the suggested applications of the new coating, yet this topic is not covered in the introduction. Adding some information on the dental implants would also help to justify the choice of cells for the biocompatibility assays (in particular, the DPSCs).

Apart from that, I would advise to support the statistical data (lines 45-49) with references. In lines 50-51, the sentence “Frequent infection origin…” could be rephrased for greater clarity.

In my opinion, the introduction somewhat downplays the significance of this particular study. It was not really clear to me what the main achievement was: the manganese doping? Being able to prepare this type of coating with the arc plasma deposition? Perhaps, the focus of the study should be defined a bit better to help the reader see the reasoning behind the experiments summarized in the last paragraph.

The materials and methods section provides a clear overview of the procedures used in the study, with one exception. In lines 133-136, the authors describe the preparation of extracts for the cytotoxicity assay, referencing ISO 10993.5-2011. To my knowledge, this standard allows for a range of conditions during the extraction. In this case, it would be more helpful for the readers to have a complete description of the actual procedure in the manuscript itself.

In some cases, the description of the cell culture protocols has a somewhat unusual terminology. If I am not mistaken, in lines 148, 149, 161, “tablet” should be replaced by “plate”; in lines 148, 149, 161, “hole” should be replaced by “well”; in lines 154-155, “sown” and “sowing” should be replaced by “seeded” and “seeding”; in line 164, “colored” should be replaced by “stained”, and in line 170, “Cup” should be replaced by “dish”. There might also be a formatting error in line 169, with “106 CFU” probably being meant as “106 CFU”.

In the results and discussion section, I found certain figures to be somewhat misleading. In figure 6 (A, C), it is difficult to compare the gross morphology of the TCP and the Mn-TCP coatings. I would recommend using the SEM images of the same magnification.

In figure 7 (B), the morphology of the rosettes formed on the surface seems to be only marginally similar to that shown in the SEM images in ref. 31. The size and the “sharpness” of the crystals look quite different to me – could the authors comment on that? It would also be interesting to compare the surface changes of the soaked Mn-TCP coating to those of the control TCP coating – does the same process of OCP formation take place there, and if yes, do the crystals look the same? Additionally, in the accompanying text, the abbreviation “OCP” is introduced before the term “octacalcium phosphate”, making the description less clear.

Figure 8 presents the contact angles of the coated and non-coated surfaces. The y-axis is labeled as “Contact angle, %”, however, it looks like the actual contact angles are plotted for TCP and Mn-TCP coating. If this is the case, the y-axis label and the control bar should be changed, since the contact angle for clear glass and distilled water is around 50°. Alternatively, the TCP and the Mn-TCP bars should be altered to reflect the fact that both coatings are more hydrophobic than glass (control).

In lines 301-304 and in figure 9, the authors describe the results of the MTT assay in terms of the “survivability” of the cells. However, MTT assay measures the metabolic activity of the cells, which is, of course, affected by the number of viable cells, but is not limited to that. I would hence suggest replacing “survivability” with “metabolic activity”.

In the section on DPSCs, the results of the resazurin assay are not shown in the manuscript, which makes it hard to assess the discussion of those data.

In the paragraph describing the SEM images (lines 340-346, fig. 12), the authors do not mention the timepoint, at which the images were taken. This makes it hard to correlate the SEM data with the results shown in fig. 11. The larger number of cells on the Mn-TCP coating, described in lines 343-345, is not really clear from fig. 12 and is not supported by the data shown in fig.11, where the differences between two coatings are not very pronounced. Moreover, the differences in the cell counts on the two coatings could be explained by the lower hydrophobicity of Mn-TCP, resulting in a larger number of cells initially attaching to the surface. In my opinion, additional experiments would be necessary to distinguish between the effects of the physical properties and the chemical composition of the coating.

For the assays concerning the inhibition of bacterial growth, I would suggest testing the growth of bacteria on the coated surfaces, rather than looking for the inhibition zones. If I understand correctly, in ref.8, the Mn-TCP powder was in direct contact with the bacterial cells, as it was dissolved in the culture medium. Moreover, to my knowledge, most of the implant-associated infections are associated with the bacterial growth on the actual surface of the implants. If the material described in the current manuscript retains the same contact antibacterial properties as the powder, it would be an interesting and important feature for a potential implant coating. However, this cannot be tested with the inhibition zone assay, which reflects the cytotoxic and cytostatic properties of the substances diffusing from the sample.

Author Response

please, see the attachment

Round 2

Reviewer 1 Report

Recommend to publish. Only one minor issue

In Sentense " To our knowledge, author [13] are the first to report......" SHOULD BE

"To our knowledge, Sima et al. are the first to report............[13] "

I mean "author" in my first round of comment is to give the name of author. Not directly write "author"